# STRATEGIC REASONING IN LARGE LANGUAGE MODELS

**Ivan Selivanov**
Department of Artificial Intelligence
Kuban State University

**Anna Kovalenko**
Faculty of Computer Technology and Applied Mathematics
Kuban State University

## ABSTRACT

Large Language Models (LLMs) are increasingly deployed in settings where success depends on strategic reasoning under incentives, hidden information, and interaction with other agents. We present an empirical benchmark of strategic reasoning across nine frontier models: GPT-5.4, Claude Opus 4.6, Claude 4.6 Sonnet, Gemini 3.1 Pro, Gemini 3.1 Flash Lite, DeepSeek V3.2, Grok 4.2, Kimi K2.5, and GLM-5. The evaluation covers four canonical games spanning complete and incomplete information: Prisoner's Dilemma, Ultimatum Game, Kuhn Poker, and Texas Hold'em, under two conditions: without additional context and with additional context. We score both decision quality and reasoning quality, and aggregate them into a Strategic Reasoning Index (SRI). The results show strong heterogeneity across models: several systems are reliable on complete-information games, whereas Texas Hold'em arithmetic and multi-street decision problems remain the main stress test; additional context can improve execution for some models while compressing or distorting reasoning in others. These findings support a more diagnostic view of strategic competence and highlight the importance of output reliability in action-taking pipelines.

**Keywords.** Strategic reasoning; large language models; game theory; benchmarking; incomplete-information games; poker decision making; context effects; reasoning evaluation; multi-agent systems; game-theoretic analysis.

**TL;DR.** An empirical benchmark evaluating frontier LLMs on game-theoretic reasoning in complete and incomplete information games, with and without additional context.

## 1 INTRODUCTION

### 1.1 RESEARCH FRAMEWORK

Large language models increasingly operate in settings where success depends not only on factual knowledge or local reasoning, but on incentives, hidden information, and expectations about other agents. This makes game-theoretic evaluation a useful lens for studying strategic reasoning: whether a model can map a formal interaction to an action that is defensible under the relevant solution concept (Osborne & Rubinstein, 1994; Fudenberg & Tirole, 1991). Existing work has already shown that LLMs can be benchmarked on games (Duan et al., 2024; Wang et al., 2024; Costarelli et al., 2024; Huang et al., 2025) and that context can alter model behavior (Lorè & Heydari, 2024; Li et al., 2025). The unresolved issue for this study is how frontier models differ in strategic profiles across canonical games, and whether additional context changes their decisions in a consistent and interpretable way.

This study evaluates nine frontier models on four games spanning complete- and incomplete-information settings: Prisoner's Dilemma, Ultimatum Game, Kuhn Poker, and Texas Hold'em. Each model is tested in two conditions: *without additional context* and *with additional context*. In this paper, additional context refers to controlled supporting material such as compact theory reminders and solution-method cues, rather than unconstrained browsing. This design lets us compare the same models on the same tasks while isolating the role of external conceptual support.

Table 1: Study at a glance

| Component | Specification |
|---|---|
| Models | GPT-5.4; Claude Opus 4.6; Claude 4.6 Sonnet; Gemini 3.1 Pro; Gemini 3.1 Flash Lite; DeepSeek V3.2; Grok 4.2; Kimi K2.5; GLM-5 |
| Games | PD; UG; Kuhn Poker; Texas Hold'em |
| Conditions | without context; with context |
| Metrics | Decision Quality; Reasoning Quality; SRI |

We evaluate each response along two axes. **Decision Quality** (DQ) measures whether the selected action is strategically correct or close to the benchmark-optimal answer. **Reasoning Quality** (RQ) measures whether the model's public justification reflects the right strategic logic. These dimensions are aggregated into the **Strategic Reasoning Index** (SRI), which allows us to compare action quality and explanation reliability across models, games, and prompt conditions.

Our study is organized around three research questions:

1. How closely do frontier LLMs approximate benchmark-optimal strategies across complete- and incomplete-information games?
2. What systematic strategic profiles differentiate models across games and information regimes?
3. How does additional context affect decision quality and public justification quality?

## 1.2 GAME SELECTION & TASK DESIGN

We selected four games to balance theoretical clarity with strategic diversity. Prisoner's Dilemma (Osborne & Rubinstein, 1994) and Ultimatum Game (Güth et al., 1982) provide analytically transparent complete-information settings with well-defined equilibrium structure, while Kuhn Poker (Kuhn, 1950) and Texas Hold'em (Chen & Ankenman, 2006; Bowling et al., 2015) introduce hidden information, belief-dependent play, and sequential decision making.

The benchmark is organized around two families. The complete-information games diagnose dominance reasoning, Nash and subgame-perfect equilibrium logic (Osborne & Rubinstein, 1994), and acceptance-threshold computation (Fehr & Schmidt, 1999). The incomplete-information games probe information-set reasoning, mixed-strategy consistency (Kuhn, 1950), pot-odds and expected-value calculations (Chen & Ankenman, 2006), and the ability to act under uncertainty rather than from a fully observed payoff matrix.

In total, the study uses eighteen tasks distributed across the four games. Rather than repeating near-identical variants, the task set was designed around representative strategic structures: one-shot equilibrium analysis, utility modification, finite repetition and grim-trigger incentives in PD; proposer and responder cases, discrete and continuous bargaining, and Fehr–Schmidt thresholds (Fehr & Schmidt, 1999) in UG; dominated-action screens, equilibrium bluff and call frequencies, and game value in Kuhn Poker; and pot-odds, two-street expected value, and jam-or-fold approximation in Texas Hold'em. In the contextual condition, models receive controlled supporting material such as compact theory reminders and solution-method cues, but not unconstrained browsing and not verbatim leakage of benchmark answers.

## 1.3 EXPERIMENTAL PROTOCOL & METRICS

The study evaluates nine frontier models: GPT-5.4, Claude Opus 4.6, Claude 4.6 Sonnet, Gemini 3.1 Pro, Gemini 3.1 Flash Lite, DeepSeek V3.2, Grok 4.2, Kimi K2.5, and GLM-5. All models are tested in their standard thinking mode through provider web-based chat interfaces, without additional system-level tweaks beyond shared output-format instructions. Each model is evaluated in two conditions. The task battery is identical across conditions, and each run is issued as a single unified prompt requiring a structured JSON answer. The contextual condition uses a frozen block of supporting material rather than live web access, so that the intervention remains comparable across models and reproducible across runs.

Table 2: Game families and task design

| Game | Info regime | Core skill | Task types |
|---|---|---|---|
| PD | Complete | dominance, NE, repetition | one-shot NE, modified utility, finite rep., grim trigger |
| UG | Complete | SPE, thresholds | proposer/responder, discrete/ continuous, Fehr–Schmidt |
| Kuhn Poker | Incomplete | info sets, mixed strategies | dominated-action screens, bluff/call freq., game value |
| Texas Hold'em | Incomplete | sequential EV | pot odds, two-street EV, jam-or-fold |

Evaluation combines task-level correctness with rubric-based scoring on two public dimensions: DQ and RQ. These are aggregated into the SRI, which serves as the model-level summary measure. The exact no-context prompt, the frozen reference-context packet, the run log, and the per-task scoring sheets are archived in the supplementary artifact package so that the reported SRI values can be recomputed from released run-level data. To complement the strategic evaluation, we also track external operational metadata: response speed is taken from Artificial Analysis, and cost information is taken from official provider pricing pages rather than from chat-interface telemetry. Because the benchmark contains one run per model-condition pair on a compact 18-task battery, we interpret differences descriptively rather than as inferential significance claims.

## 2 RELATED WORK

**Game-theoretic benchmarks for LLMs.** The intersection of LLMs and game theory has grown rapidly; Sun et al. (2025) provide a comprehensive survey. On the benchmark side, GTBench (Duan et al., 2024) introduced a systematic evaluation suite covering ten games including Kuhn Poker and iterated Prisoner's Dilemma, revealing that even strong LLMs exhibit significant strategic failures. TMGBench (Wang et al., 2024) expanded coverage to 144 types of $2 \times 2$ games, testing robustness and theory-of-mind reasoning at scale. GameBench (Costarelli et al., 2024) broadened the evaluation environments further, reinforcing that raw LLM strategic play is limited. GAMA-Bench (Huang et al., 2025) extended the paradigm to multi-agent settings. These benchmarks establish that games-as-evaluation is a productive paradigm (Shoham & Leyton-Brown, 2008), but they generally report aggregate accuracy without separating decision quality from the quality of the underlying strategic justification. Our benchmark is deliberately narrower: four canonical games, nine models, and a scoring rubric that disentangles action correctness from reasoning depth.

**Behavioral games and LLMs.** A separate line of work evaluates LLMs on behavioral games originally designed for human experiments. Mei et al. (2024) found that base LLMs exhibit human-like reasoning biases, including cooperation in PD and fair offers in UG, that largely disappear in fine-tuned chatbots. Fontana et al. (2024) showed that LLMs cooperate more readily than humans in iterated PD, while Gawne et al. (2024) studied single- vs. multi-agent LLM configurations in the Ultimatum Game. Xie et al. (2024) compared chatbot behavior across several behavioral games. Willis et al. (2025) demonstrated that strategic prompting can shift cooperative dynamics in iterated PD. These studies confirm that LLMs produce meaningful, and often human-like, strategic behavior, but they typically evaluate decisions alone without scoring the accompanying reasoning.

**Poker and incomplete information.** Poker provides a natural incomplete-information testbed with a well-developed computational tradition (Burch et al., 2014; Bowling et al., 2015), and AI systems have achieved human-level play in Diplomacy by combining language models with strategic reasoning (Meta Fundamental AI Research Diplomacy Team (FAIR), 2022). More recently, Poker-Bench (Zhuang et al., 2025) introduced a benchmark for Hold'em spots and showed that LLMs struggle with sequential EV computation. ToolPoker (Lin et al., 2026) demonstrated that solver-assisted tool use can compensate for raw LLM limitations. PokerGPT (Huang et al., 2024) explored end-to-end LLM poker play, and Qin et al. (2025) studied collusion in three-player Kuhn Poker

Table 3: Complete-information games. Values are reported as no-context / with-context. RQ denotes average reasoning-quality score. Speed comes from Artificial Analysis. Output price uses official provider pricing; * marks the closest public proxy for Grok 4.2.

| Model | PD acc. | UG acc. | PD RQ | UG RQ | t/s | $/1M |
|---|---|---|---|---|---|---|
| GPT-5.4 | 1.00 / 1.00 | 1.00 / 1.00 | 3.00 / 3.00 | 3.00 / 3.00 | 72.5 | 15.00 |
| Claude Opus 4.6 | 1.00 / 1.00 | 1.00 / 1.00 | 3.00 / 3.00 | 3.00 / 3.00 | 44.1 | 25.00 |
| Claude 4.6 Son. | .75 / 1.00 | .75 / 1.00 | 2.75 / 2.25 | 2.75 / 2.75 | 57.3 | 15.00 |
| Gemini 3.1 Pro | 1.00 / 1.00 | 1.00 / 1.00 | 3.00 / 3.00 | 3.00 / 3.00 | 100.4 | 12.00 |
| Gem. 3.1 Fl. Lt. | .75 / 1.00 | 1.00 / 1.00 | 2.25 / 3.00 | 3.00 / 3.00 | 316.5 | 1.50 |
| DeepSeek V3.2 | 1.00 / 1.00 | 1.00 / 1.00 | 2.25 / 2.25 | 2.75 / 2.75 | 27.2 | 0.42 |
| Grok 4.2 | 1.00 / .75 | 1.00 / 1.00 | 2.25 / 2.00 | 2.75 / 2.75 | 40.8 | 15.00* |
| Kimi K2.5 | .75 / .75 | .50 / 1.00 | 2.50 / 2.50 | 2.25 / 2.75 | 45.3 | 3.00 |
| GLM-5 | 1.00 / 1.00 | 1.00 / 1.00 | 3.00 / 2.25 | 2.75 / 2.75 | 60.0 | 3.20 |

through cheap talk. Our benchmark includes both Kuhn Poker and Texas Hold'em but focuses on whether models can reconstruct the strategic logic behind correct actions, not just produce them.

**Context and framing effects.** Lorè & Heydari (2024) showed that contextual framing can alter LLM strategic behavior even when the formal game structure is unchanged. Li et al. (2025) reinforced this finding across broader decision tasks, and Guo et al. (2024) documented similar effects in auction and beauty-contest settings. Jia et al. (2025) connected LLM strategic evaluation to behavioral game theory, highlighting the role of context in shaping model responses. Our study builds on these findings by comparing a specific type of context, namely a frozen packet of compact theory reminders and solution-method cues, against a no-context baseline across nine models and four games. The central issue is how this contextual support interacts with each model's latent strategic competence.

## 3 BENCHMARK

### 3.1 COMPLETE INFORMATION GAMES

Prisoner's Dilemma and Ultimatum Game provide the cleanest complete-information tests in the benchmark. In these settings, the relevant solution concepts are analytically transparent: dominance and backward-induction logic in PD (Osborne & Rubinstein, 1994), and proposer–responder threshold structure in UG (Güth et al., 1982; Fehr & Schmidt, 1999). This makes the complete-information block useful not only for measuring whether models can recover a correct strategic answer, but also for testing whether they can serialize that reasoning cleanly into a stable final output.

Across the nine-model roster, stronger systems are generally reliable on complete-information tasks, but the interesting variation appears one layer deeper. The sharper distinction is between models that keep their final structured answer aligned with their reasoning and models that lose that alignment during serialization.

**Example 1. Complete-information self-correction without integration.** Task UG-3 asks for the responder threshold under Fehr–Schmidt inequality aversion. Kimi K2.5 writes `1/2` in the structured answer, but the same response later derives $3s - 1 \geq 0$, correctly concluding that the threshold is $1/3$. The model reaches the correct strategic logic but fails to propagate that correction into the scored answer field.

### 3.2 INCOMPLETE INFORMATION GAMES

Kuhn Poker and Texas Hold'em provide the incomplete-information half of the benchmark, but they stress different capabilities. Kuhn Poker (Kuhn, 1950) is small and analytically tractable, so it isolates information sets, mixed-strategy consistency, and equilibrium facts in a setting where the strategic target is explicit. Texas Hold'em is structurally harder (Chen & Ankenman, 2006; Bowling et al., 2015): it requires local EV calculations, correct interpretation of future cards and betting opportunities, and reliable action selection under hidden information.

Table 4: Incomplete-information games. Notation follows Table 3.

| Model | KP acc. | TH acc. | KP RQ | TH RQ | t/s | $/1M |
|---|---|---|---|---|---|---|
| GPT-5.4 | .75 / 1.00 | .83 / .50 | 1.75 / 2.00 | 2.67 / 1.50 | 72.5 | 15.00 |
| Claude Opus 4.6 | 1.00 / 1.00 | .83 / .83 | 2.75 / 2.00 | 2.67 / 2.67 | 44.1 | 25.00 |
| Claude 4.6 Son. | 1.00 / 1.00 | .33 / .83 | 2.75 / 2.00 | 2.00 / 2.17 | 57.3 | 15.00 |
| Gemini 3.1 Pro | 1.00 / 1.00 | .67 / .67 | 2.75 / 2.00 | 2.50 / 2.33 | 100.4 | 12.00 |
| Gem. 3.1 Fl. Lt. | 1.00 / 1.00 | .67 / .67 | 2.25 / 2.00 | 1.50 / 1.67 | 316.5 | 1.50 |
| DeepSeek V3.2 | 1.00 / 1.00 | 1.00 / .67 | 2.00 / 2.00 | 2.83 / 2.00 | 27.2 | 0.42 |
| Grok 4.2 | 1.00 / 1.00 | .50 / .83 | 2.00 / 2.00 | 1.83 / 2.17 | 40.8 | 15.00* |
| Kimi K2.5 | .75 / 1.00 | .00 / .33 | 1.75 / 2.00 | 1.50 / 1.83 | 45.3 | 3.00 |
| GLM-5 | 1.00 / 1.00 | .67 / 1.00 | 2.00 / 2.00 | 2.33 / 2.50 | 60.0 | 3.20 |

Table 5: Context effect by model

| Model | $\Delta$acc. (pp) | $\Delta$SRI | Context effect type |
|---|---|---|---|
| GPT-5.4 | −5.6 | −9.26 | context-fragility on multi-street EV |
| Claude Opus 4.6 | +0.0 | −2.78 | stable; explanation compression |
| Claude 4.6 Sonnet | +27.8 | +5.56 | strong positive regularization |
| Gemini 3.1 Pro | +0.0 | −4.63 | retrieval-like compression |
| Gem. 3.1 Fl. Lt. | +5.6 | +3.71 | compressed, improved by context |
| DeepSeek V3.2 | −11.1 | −8.34 | heuristic overbinding |
| Grok 4.2 | +5.6 | +0.93 | regularization w/o depth gain |
| Kimi K2.5 | +27.8 | +9.26 | self-correction assist |
| GLM-5 | +11.1 | +2.77 | context-regularized exec. gain |

The empirical split across models is correspondingly sharp. Many models remain solid on Kuhn Poker once the equilibrium structure is aligned, but Texas Hold'em becomes the main separator. Here the dominant failure modes are not simply "wrong arithmetic"; they include one-card versus two-card misreadings, heuristic overbinding under context, approximation-heavy execution, and unstable conversion of local calculations into final actions.

**Example 2.   The same poker task, two different context effects.** Task TH-3 is a flop decision with eight clean outs and no future betting. GLM-5 with context treats the spot as a two-card realization and calls profitably. DeepSeek V3.2 with context collapses to a one-card heuristic reading and folds. The same contextual support can regularize one model while overbinding another to a narrower decision rule.

### 3.3   CONTEXT EFFECT ANALYSIS

Comparing each model with itself shows that context effects are heterogeneous rather than uniformly beneficial, consistent with prior findings on framing sensitivity (Lorè & Heydari, 2024; Li et al., 2025; Guo et al., 2024). In this benchmark, additional context can preserve strong performance, regularize unstable execution, compress independent reasoning into retrieval-like justification, or induce heuristic overbinding. What matters here is how context changes the relation between strategic competence and answer reliability.

## 4   RESULTS & CONCLUSIONS

### 4.1   PERFORMANCE ANALYSIS

The central metric in this study is the **Strategic Reasoning Index** (SRI), a composite score that combines action quality and public justification quality across the full task battery. For a model evaluated on $N$ tasks,

$$\text{SRI} = \frac{1}{6N} \sum_{i=1}^{N} (D_i + R_i) \times 100,$$

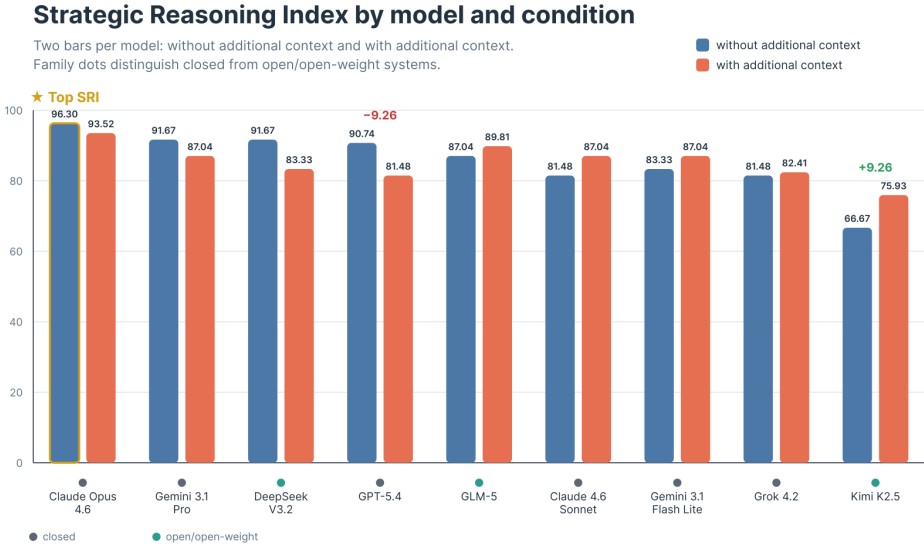

Figure 1: Strategic Reasoning Index by model and condition. The highest absolute SRI belongs to Claude Opus 4.6, the largest positive context shift belongs to Kimi K2.5, and the largest negative shift belongs to GPT-5.4.

Table 6: Overall model comparison. Sorted by best observed SRI. Fam. abbreviations: Cl.=closed, Op.=open/open-weight. $^*$ marks Grok 4 proxy pricing.

| Model | Fam. | $SRI_{nc}$ | $SRI_{wc}$ | Best | t/s | $/1M |
|---|---|---|---|---|---|---|
| Claude Opus 4.6 | Cl. | 96.30 | 93.52 | 96.30 | 44.1 | 25.00 |
| Gemini 3.1 Pro | Cl. | 91.67 | 87.04 | 91.67 | 100.4 | 12.00 |
| DeepSeek V3.2 | Op. | 91.67 | 83.33 | 91.67 | 27.2 | 0.42 |
| GPT-5.4 | Cl. | 90.74 | 81.48 | 90.74 | 72.5 | 15.00 |
| GLM-5 | Op. | 87.04 | 89.81 | 89.81 | 60.0 | 3.20 |
| Claude 4.6 Son. | Cl. | 81.48 | 87.04 | 87.04 | 57.3 | 15.00 |
| Gem. 3.1 Fl. Lt. | Cl. | 83.33 | 87.04 | 87.04 | 316.5 | 1.50 |
| Grok 4.2 | Cl. | 81.48 | 82.41 | 82.41 | 40.8 | 15.00$^*$ |
| Kimi K2.5 | Op. | 66.67 | 75.93 | 75.93 | 45.3 | 3.00 |

where $D_i$ and $R_i$ denote the Decision Quality and Reasoning Quality scores for task $i$. Plain accuracy alone cannot distinguish between a model that reaches the correct answer through stable strategic reasoning and one that arrives there with shallow or unstable justification.

Figure 1 highlights three patterns. Claude Opus 4.6, Gemini 3.1 Pro, and DeepSeek V3.2 form the strongest no-context tier. The models with the highest raw performance are not the ones with the largest contextual gains: Kimi K2.5 and Claude 4.6 Sonnet improve sharply under context, while GPT-5.4 and DeepSeek V3.2 decline. GLM-5 and Gemini 3.1 Flash Lite both improve in execution under context, but in distinct ways: GLM-5 remains stable and balanced, whereas Flash Lite stays compressed and explanation-thin.

SRI is therefore only the first axis of comparison. For deployment, speed and output price matter as well, because repeated model calls compound latency and cost in agentic settings.

The closed-family contextual mean stays comparatively strong, but with much larger within-family variance than the open/open-weight group.

Table 7: Game-level aggregate performance

| Game | Acc.$_{nc}$ | Acc.$_{wc}$ | SRI$_{nc}$ | SRI$_{wc}$ |
|---|---|---|---|---|
| Prisoner's Dilemma | .917 | .944 | 92.13 | 90.28 |
| Ultimatum Game | .917 | 1.000 | 94.91 | 97.68 |
| Kuhn Poker | .944 | 1.000 | 85.18 | 83.33 |
| Texas Hold'em | .611 | .704 | 74.69 | 75.31 |

Table 8: Closed vs. open/open-weight comparison. Med. denotes median.

| Family | $n$ | SRI$_{nc}$ | SRI$_{wc}$ | $\Delta$SRI | Med. \$/1M |
|---|---|---|---|---|---|
| Closed | 6 | 87.50 | 86.42 | $-1.08$ | 15.00 |
| Open/open-wt. | 3 | 81.79 | 83.02 | $+1.23$ | 3.00 |

## 4.2 FAILURE MODES

The error profile of the benchmark is not homogeneous, echoing findings from behavioral game theory that strategic failure in both humans and LLMs is multi-dimensional (Camerer, 2003; Mei et al., 2024; Jia et al., 2025). Some mistakes are simple wrong-answer cases, but others reflect a deeper mismatch between latent strategic competence and scored output. Across the full matrix, six failure modes recur (Table 9).

**Example 3.   Execution fails after the right reasoning appears.** Task TH-1 in Claude 4.6 Sonnet (no context): the structured answer says `fold`, but the public reasoning later recomputes the spot and correctly concludes that `call` is profitable. Strategic reasoning is present, but execution fails at the final answer layer.

**Example 4.   Context improves execution without deepening the reasoning style.** Tasks TH-4 and TH-R1 in Gemini 3.1 Flash Lite (with context): the model now produces usable answers, but its reasoning remains compressed and threshold-heavy. Context can regularize a lightweight model operationally without turning it into a deeper strategic explainer.

## 4.3 IMPLICATIONS & CONCLUSION

**RQ1** asked how closely frontier LLMs approximate benchmark-optimal strategies. Consistent with prior observations (Duan et al., 2024; Wang et al., 2024), the answer is: often quite closely, but unevenly. Several models are highly reliable on complete-information games and on Kuhn Poker, while Texas Hold'em remains the main stress test for pot-odds arithmetic and multi-street interpretation.

**RQ2** asked whether models exhibit systematic strategic profiles. They do. The current matrix suggests a working taxonomy: Claude Opus 4.6 behaves as a *deep strategic explicator*; DeepSeek V3.2 as a *sparse but stable executor*; Gemini 3.1 Pro as a *formal-template strategist*; Gemini 3.1 Flash Lite as a *compressed threshold strategist*; Grok 4.2 as an *approximate strategist regularized by context*; Kimi K2.5 as a *self-correcting but weakly integrated strategist*; GLM-5 as a *formal and execution-stable strategist*; and Claude 4.6 Sonnet under context as a case of *strong positive regularization*.

**RQ3** asked how additional context changes decision quality and justification quality. The effect is not uniform. Context can preserve high performance while compressing explanation depth, regularize unstable execution, induce heuristic overbinding, or improve a compressed model without making it substantially deeper. In practice, model selection is not just a matter of maximizing SRI. The relevant deployment choice is the balance between strategic quality, speed, and price: Claude Opus 4.6 leads on quality, but GLM-5 and DeepSeek V3.2 offer much stronger efficiency, while Gemini 3.1 Flash Lite shows that low-cost speed can be operationally useful if the contextual support is well aligned.

One implication for multi-agent systems and other action-taking pipelines is straightforward: output reliability matters as much as latent strategic competence (Wang et al., 2026; Bianchi et al.,

Table 9: Error taxonomy

| Failure mode | Models | Tasks |
|---|---|---|
| Answer-finalization instab. | Claude 4.6 Son. (nc) | PD-2, TH-1, TH-4 |
| Self-correction integ. fail | Kimi K2.5 | UG-3, TH-1, TH-3 |
| Heuristic overbinding | DeepSeek V3.2 (wc) | TH-3 |
| Compressed threshold reas. | Gem. 3.1 Fl. Lt. | TH-4, TH-R1 |
| Approximation-heavy exec. | Grok 4.2 | TH-2, TH-R1 |
| Benchmark-cond. retrieval | Opus / GLM / Pro (wc) | KP-2–4 |

2024; Hua et al., 2024). A model that reasons well but serializes poorly remains risky in negotiation, coordination, and adversarial settings (Fish et al., 2024), because downstream agents only see the final action. We treat this as an external implication of the benchmark rather than as a direct multi-agent evaluation result. This study remains limited by its manual web-chat protocol, fixed contextual packet, external speed and pricing metadata, compact 18-task battery, descriptive rather than inferential analysis, absence of a reported formal inter-rater coefficient, and restriction to four games and one style of contextual intervention. Contamination risk is reduced by adapting rather than copying most benchmark items, but canonical Kuhn Poker facts remain inherently more exposed than the constructed PD, UG, and Hold'em arithmetic tasks. Future work could extend the benchmark to broader game families (Huang et al., 2025; Wang et al., 2024), integrate solver-backed poker evaluation (Zhuang et al., 2025; Lin et al., 2026), report formal inter-rater agreement, or study multi-agent negotiation dynamics (Bianchi et al., 2024; Hua et al., 2024). The benchmark nevertheless shows that strategic reasoning in LLMs is not a single scalar trait, but a multi-dimensional and context-sensitive capability shaped by how models connect formal reasoning, local interpretation, and final action.

## AUTHOR CONTRIBUTIONS

Ivan Selivanov designed the benchmark, conducted all experiments, performed scoring and analysis, and drafted the manuscript. Anna Kovalenko supervised the research design, verified the scoring rubric, and revised the manuscript.

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

## A  PROMPT TEMPLATES

### A.1  NO-CONTEXT PROMPT

The no-context prompt contains no external reference material. The full no-context prompt, including all 18 task descriptions, is archived in the supplementary materials. Its instruction header begins with:

```
You are participating in a controlled research test of
strategic reasoning.

Rules:
- Solve all tasks in one response.
- Use only the information given in this prompt.
- Do not use external knowledge retrieval, web browsing,
tool use, or hidden assumptions.
- Return only valid JSON.

Condition notice:  This run is under the condition:
no_context.

Reference context:  No external reference context is
provided for this test.
```

### A.2  WITH-CONTEXT REFERENCE BLOCK

The with-context prompt prepends a frozen reference-context packet before the identical task battery. This packet provides compact theory reminders and solution-method cues but does not include verbatim benchmark answers. It covers core definitions (dominant strategy, Nash equilibrium, SPE, information sets, mixed strategies), game-specific theory for PD, UG, Kuhn Poker, and Texas Hold'em, and closes with: "Use these principles only as solution methods. Do not invent facts not stated in the task itself." The full with-context prompt and the exact frozen packet are archived in the supplementary materials.

### A.3  OUTPUT SCHEMA

All models receive identical output requirements: exactly 18 answer objects in a single JSON, each containing `task_id`, `structured_answer` (with task-specific keys), `final_answer_short`, `reasoning` ($\leq$120 words), and `confidence` $\in [0, 1]$. The released run-level files preserve these fields for all 18 model-condition runs.

## B  BENCHMARK INSTANCES AND ANSWER KEY

Table 10 summarizes the strategic coverage of the 18-task battery. The supplementary materials contain the full verbatim task texts; the appendix here records the compact task inventory and the scoring key used for manual evaluation.

The Hold'em tasks are intentionally hand-computable EV probes rather than solver-backed full-range analyses. TH-2 uses one-card pot-odds arithmetic on the turn, TH-3 uses two-card realization

Table 10: Task battery summary

| Family | Tasks | Strategic focus |
|---|---|---|
| PD | PD-1–4 | one-shot dominance/NE; utility modification; finite repetition; grim-trigger threshold |
| UG | UG-1–4 | proposer optimum; discrete responder threshold; continuous cutoff; Fehr–Schmidt offer |
| KP | KP-1–4 | dominated-action screens; bluff frequency; call frequency; game value |
| TH | TH-1–5, TH-R1 | river pot odds; turn EV; flop two-card EV; sequential updates; jam-or-fold approximation |

Table 11: Ground-truth answers

| Task | Key field(s) | Ground truth |
|---|---|---|
| PD-1 | dom. strategies; NE; Pareto | D, D; (D,D); (C,C) |
| PD-2 | alpha condition | $\alpha < 1/2$ |
| PD-3 | SPE path | (D,D) all 3 periods |
| PD-4 | delta threshold | $\delta \geq 1/2$ |
| UG-1 | offer share | 0 |
| UG-2 | offer integer | 1 |
| UG-3 | acceptance threshold | 1/3 |
| UG-4 | optimal offer share | 0.3 |
| KP-1 | weakest cont.; strongest fold | No; No |
| KP-2 | bluff frequency | 1/3 |
| KP-3 | call frequency | 1/3 |
| KP-4 | game value | $-1/18$ |
| TH-1 | river action | Call |
| TH-2 | turn action; EV | Call; $\approx 46.09$ |
| TH-3 | flop action; EV | Call; $\approx 17.18$ |
| TH-4 | flop; turn | Call; Call |
| TH-5 | flop; turn | Call; Fold |
| TH-R1 | jam frequency | $\approx 61.7\%$ |

from the flop under an explicit no-future-betting assumption, and TH-R1 uses a fixed benchmark jam frequency from the released answer key.

Acceptance rules: fractions, decimals, and percentages are treated as equivalent when numerically equal. For TH-2 and TH-3, close positive approximations to the listed EVs are accepted. When `final_answer_short` conflicts with `structured_answer`, the scorer treats `structured_answer` as the primary field and records the inconsistency.

## C  SCORING RUBRIC

**Decision Quality (DQ).**   Scored 0–3: (0) no valid decision; (1) strongly suboptimal action; (2) partially correct; (3) optimal or near-optimal. For Kuhn Poker, DQ = 3 requires probability sum in $[0.98, 1.02]$, off-support mass $\leq 0.05$, and $L_1$ distance to reference $\leq 0.20$.

**Reasoning Quality (RQ).**   Scored 0–3: (0) no relevant reasoning; (1) superficial or conceptually wrong; (2) competent but incomplete; (3) rigorous and complete. One critical logical error disqualifies score 3.

**Composite metric.**   $\text{SRI} = \frac{1}{6N} \sum_i (D_i + R_i) \times 100$, where $N$ is the number of tasks. Sub-indices SRI-Complete (PD + UG) and SRI-Incomplete (KP + TH) are computed over the relevant subsets.

**Annotation protocol.** Two annotators independently score a calibration set ($\geq 10$ outputs) before the main batch. The main batch is blinded (randomized order, no model identity). DQ is scored first, then RQ separately. Disagreements $>1$ point go to adjudication.

The scoring manual, run log, and per-task DQ/RQ sheets for all 18 runs are released in the supplementary materials, and the main-text SRI values are computed from those run-level tables. The annotation workflow therefore emphasizes transparency and replayability, but we do not report a formal inter-rater agreement coefficient in the current version.

## D  MODEL FREEZE SHEET

All experiments were conducted on March 6, 2026. Each model was tested in its standard thinking/reasoning mode via the provider's web-based chat interface.

Table 12: Model identification

| Model | Provider | Mode | Date |
|---|---|---|---|
| GPT-5.4 | OpenAI | Thinking (xhigh) | 2026-03-06 |
| Claude Opus 4.6 | Anthropic | Adaptive (Max) | 2026-03-06 |
| Claude 4.6 Sonnet | Anthropic | Adaptive (Max) | 2026-03-06 |
| Gemini 3.1 Pro | Google DM | Thinking (Preview) | 2026-03-06 |
| Gem. 3.1 Fl. Lt. | Google DM | Thinking (Preview) | 2026-03-06 |
| DeepSeek V3.2 | DeepSeek | Reasoning | 2026-03-06 |
| Grok 4.2 | xAI | Thinking (beta) | 2026-03-06 |
| Kimi K2.5 | Moonshot AI | Thinking | 2026-03-06 |
| GLM-5 | Zhipu AI | Thinking | 2026-03-06 |

Table 13: Official pricing (as of 2026-03-06). $^*$ Grok 4 stable proxy.

| Model | Input \$/1M | Output \$/1M |
|---|---|---|
| GPT-5.4 | 2.50 | 15.00 |
| Claude Opus 4.6 | 5.00 | 25.00 |
| Claude 4.6 Sonnet | 3.00 | 15.00 |
| Gemini 3.1 Pro | 2.00 | 12.00 |
| Gem. 3.1 Fl. Lt. | 0.50 | 1.50 |
| DeepSeek V3.2 | 0.28 | 0.42 |
| Grok 4.2 | 3.00$^*$ | 15.00$^*$ |
| Kimi K2.5 | 0.60 | 3.00 |
| GLM-5 | 1.00 | 3.20 |

Table 14: Speed and platform availability (Artificial Analysis, 2026-03-06). $^*$ Grok 4 proxy.

| Model | t/s | TTFT (s) | AA prov. | Platforms |
|---|---|---|---|---|
| GPT-5.4 | 72.5 | 185.0 | 1 | 3 |
| Claude Opus 4.6 | 44.1 | 14.9 | 4 | 6 |
| Claude 4.6 Sonnet | 57.3 | 104.6 | 4 | 5 |
| Gemini 3.1 Pro | 100.4 | 38.4 | 2 | 3 |
| Gem. 3.1 Fl. Lt. | 316.5 | 8.7 | 1 | 3 |
| DeepSeek V3.2 | 27.2 | 1.6 | 9 | 5 |
| Grok 4.2 | 40.8$^*$ | 12.2$^*$ | 2$^*$ | 0 |
| Kimi K2.5 | 45.3 | 3.0 | 11 | 6 |
| GLM-5 | 60.0 | 1.9 | 9 | 4 |

## E  QUALITATIVE CASE STUDIES

**E.1. Self-correction integration failure (Kimi K2.5).** On UG-3 (no context), Kimi writes $1/2$ in the structured answer but derives $3s - 1 \geq 0$ in its reasoning, correctly concluding $1/3$. On

TH-1, the structured answer says `fold` but the reasoning recalculates and recommends `call`. This pattern recurs across PD-2, UG-3, UG-4, TH-1, TH-3, and partially TH-5.

**E.2. Answer-finalization instability (Claude 4.6 Sonnet).** On TH-1 (no context), the structured answer says `fold` while the reasoning correctly computes pot odds favoring `call`. Under context, Sonnet's accuracy jumps from 66.7% to 94.4% (+27.8 pp), the largest positive context shift in the study.

**E.3. Heuristic overbinding (DeepSeek V3.2 with context).** On TH-3, the no-context run correctly interprets a two-card realization and calls. The with-context run binds to the cue "one card to come" and folds. Overall accuracy drops from 100% to 88.9%.

**E.4. Benchmark-conditioned retrieval.** On KP-2–4 (with context), Claude Opus 4.6, GLM-5, and Gemini 3.1 Pro produce correct answers but shift from reconstructive explanations to shorter, citation-like justifications anchored to the reference packet. KP RQ drops from $\sim$2.75 to $\sim$2.00.

**E.5. Protocol-level format failure (Claude Opus 4.6).** The no-context run (SRI $=$ 96.30, the highest in the study) appended a second JSON object, breaking the output contract. The first object is complete and correct across all 18 tasks.

## F  COST, SPEED, AND METHODOLOGY NOTES

**Pricing.** All output prices come from official provider pricing pages, frozen on March 6, 2026. Grok 4.2 was in public beta without published API pricing; Grok 4 ($3.00/$15.00) is used as proxy.

**Speed.** All speed values come from Artificial Analysis model pages (rolling 72-hour medians on first-party APIs). TTFT values are included for reference but not used in the main analysis.

**Protocol limitations.** The study uses a manual web-chat protocol rather than programmatic API calls. Token counts are not directly observable, so per-run cost cannot be computed precisely. Speed and pricing are treated as external operational metadata.

**Platform availability.** The confirmed major-platform count is based on an eight-platform checklist: own API, AWS Bedrock, Azure AI Foundry, GCP Vertex AI, Together AI, Fireworks AI, Replicate, and OpenRouter. A platform is counted only if publicly confirmed on the freeze date.

## ACKNOWLEDGEMENTS

The research was supported by a grant for organizing training for students enrolled in higher education programs for leading specialists in the field of artificial intelligence, provided by the Analytical Center under the Government of the Russian Federation under grant No. 70-2025-000735 dated May 29, 2025 (IGK 000000Ts330325R2Zh0002).

