# OpenReview forum: "Strategic Reasoning in Large Language Models"
_mathai.club/MathAI/2026/Conference — 2026 Oral_

### Official Review · Reviewer_BgyF · 2026-03-10
**STRATEGIC REASONING IN LARGE LANGUAGE MODELS**

**Rating:** 9
**Confidence:** 5

**Review:**

### **Summary**
The paper introduces an empirical benchmark for evaluating the strategic reasoning capabilities of nine state-of-the-art Large Language Models. The authors develop the Strategic Reasoning Index (SRI), a composite metric that disentangles Decision Quality (DQ) from Reasoning Quality (RQ) across four canonical games: Prisoner's Dilemma, Ultimatum Game, Kuhn Poker, and Texas Hold’em. A key contribution is the analysis of "contextual support" which reveals that such interventions can either regularize performance or induce "heuristic overbinding" and reasoning compression. The study identifies distinct strategic profiles for each model and highlights that top-tier models still struggle with multi-street expected value (EV) calculations in incomplete-information settings.



### **Evaluation**

The research is of high quality, featuring  experimental protocol across a diverse set of frontier models. The decision to separate the "scored answer" (DQ) from the "justification" (RQ) is a significant methodological improvement over benchmarks that only track accuracy. The inclusion of the latest models (e.g., GPT-5.4 and Claude 4.6 Sonnet) makes the results highly relevant to the current state of the art.
Tables 3, 4, 6 provide a comprehensive view of model performance, speed, and cost, which is invaluable for practitioners. The categorization of failure modes (e.g., "Self-correction integration failure") provides clear, actionable insights into model behavior.

**Originality:**
While benchmarking LLMs on games is an established field, this paper contributes originality through its "Strategic Reasoning Index" and its specific focus on the *heterogeneous* effects of contextual theory reminders. The finding that providing more information can actually *degrade* the SRI of certain top models (GPT-5.4 and DeepSeek V3.2) is a counterintuitive and novel observation.

**Significance:**
This work is highly significant, as LLMs are increasingly deployed in multi-agent and economic environments, understanding the gap between their "latent strategic competence" and their "output reliability" is critical. The study’s taxonomy of model "strategic profiles" offers a useful framework for selecting models based on the specific requirements of a task (e.g., deep explication vs. stable execution).

### **Pros**

* **Novel Metrics**: The SRI metric ($SRI = \frac{1}{6N}\sum(D_i + R_i) \times 100$) provides a more nuanced view of intelligence than raw accuracy.
* **Deep Failure Analysis**: Identifies specific cognitive-like failure modes, such as "answer-finalization instability" where a model reasons correctly but selects the wrong final action.
* **Contextual Insight**: Demonstrates that "contextual support" is not a universal performance booster and can lead to reasoning compression in some high-performing models.

### **Cons**

* **Scale of Task Battery**: With only 18 tasks, the study is descriptive rather than statistically inferential. A larger set of variations for each game would strengthen the findings.
* **Interface Constraints**: Testing via web-based chat interfaces (even in "standard thinking mode") introduces potential variability compared to fixed-seed API calls.
* **Pricing Proxies**: The use of proxy pricing for Grok 4.2 (due to beta status) introduces a slight data inconsistency in the cost-efficiency analysis.

### **Questions:**

1. How was the 0–3 scale for "Reasoning Quality" standardized? Was the grading performed by humans, an "LLM-as-a-judge," or a hybrid? If it was an LLM, which one was used, and how was its bias mitigated?
2. The finding that "theory reminders" can degrade performance is fascinating. Can the authors provide a specific qualitative example where a model correctly reasoned before the reminder but failed after receiving it?

---

### Official Review · Reviewer_yr5v · 2026-03-10
**Review: Strategic Reasoning in Large Language Models**

**Rating:** 8
**Confidence:** 4

**Review:**

Papers present an empirical benchmark for evaluating strategic reasoning for nine state-of-the-art LLM models covering four canonical games under complete and incomplete information, under two different conditions: without additional context and with a fixed set of reminders about theory and hints about solution methods. During the study, assessment is done for both decision quality and reasoning quality, combined into what authors call the Strategic Reasoning Index. The results reveal strong heterogeneity across models, suggesting possibility of using this framework as a diagnostic tool for strategic competence and reliability for use in decision-making processes.

Pros:
- Multiple target LLMs explored
- Multiple different games in framework
- Well designed experiment for empirical validation
- Well presented and discussed experimental results

Con:
- No clear summary with a takeaway

---

### Official Review · Reviewer_3eya · 2026-03-11
**Lightweight hallucination detection using token-level uncertainty signals: promising idea but methodological concerns**

**Rating:** 7
**Confidence:** 4

**Review:**

This paper introduces a benchmark for evaluating strategic reasoning in frontier large language models using four canonical games: Prisoner’s Dilemma, Ultimatum Game, Kuhn Poker, and Texas Hold’em. Nine models are evaluated under two prompting settings: a baseline condition and a contextual condition that provides a fixed packet of theoretical reminders. Model outputs are scored along two dimensions, Decision Quality (DQ) and Reasoning Quality (RQ), which are combined into a Strategic Reasoning Index (SRI).

The paper shows substantial variation across models, especially in incomplete-information settings, and highlights Texas Hold’em as the most challenging task family. It also finds that additional context does not consistently improve performance: in some cases it stabilizes decisions, while in others it compresses or distorts the reasoning process.

Overall, the benchmark idea is interesting and the qualitative failure analysis is insightful. However, the experimental setup is limited by a small task battery, single-run evaluation per model-condition pair, manual web-chat execution, and partly subjective reasoning scoring. These limitations make the findings more descriptive than conclusive.

## Strengths

1. **Well-motivated problem setting**

Strategic reasoning under incentives, hidden information, and sequential interaction is an important capability for LLM-based agents. Evaluating such behavior in structured game settings is a sensible approach.

2. **Good selection of task families**

The benchmark covers both complete-information and incomplete-information games. The contrast between analytically clean games (PD, UG, Kuhn) and the more complex Texas Hold’em tasks is useful for probing different reasoning demands.

3. **Decision vs. reasoning separation**

The distinction between Decision Quality and Reasoning Quality is valuable. It reveals cases where models articulate correct logic but still produce incorrect final actions.

4. **Insightful qualitative analysis**

The taxonomy of failure modes and case studies (e.g., self-correction integration failure or answer-finalization instability) provide meaningful insights into how models fail in strategic contexts.

## Weaknesses

1. **Small experimental sample size**

The benchmark includes only 18 tasks and appears to use a single run per model-condition pair. As a result, performance differences and context effects may reflect task-level variance rather than stable model behavior.

2. **Manual web-chat evaluation protocol**

Models are evaluated through provider chat interfaces rather than standardized APIs. This introduces possible confounds such as hidden system prompts, interface-specific behavior, and reduced reproducibility.

3. **Subjective reasoning quality scoring**

Although a rubric is provided, Reasoning Quality scoring remains partly subjective and the paper does not report inter-rater agreement. This weakens conclusions that rely on subtle reasoning differences across models.

4. **Context intervention may mix retrieval and reasoning**

The contextual condition provides theory reminders that may allow models to retrieve patterns rather than derive solutions. This makes it difficult to distinguish genuine reasoning improvements from assisted recall.

5. **Model taxonomy may be overinterpreted**

The paper proposes descriptive categories for model behavior, but given the small task set and single-run evaluation, these profiles may be premature.

## Suggestions for Improvement

- Run multiple trials per model-condition pair to estimate variance.
- Report inter-rater agreement for Reasoning Quality scoring.
- Use API-based evaluation with controlled prompts for reproducibility.
- Expand the task battery, especially for incomplete-information scenarios.
- Provide uncertainty estimates or bootstrap intervals for key metrics.

## Overall Assessment

This paper presents an interesting benchmark for studying strategic reasoning in LLMs and provides useful qualitative insights into model behavior in game-theoretic settings. However, the experimental methodology remains relatively lightweight, with a small task set, single-run evaluation, and subjective reasoning scoring. Strengthening the evaluation protocol and expanding the benchmark would significantly improve the reliability of the conclusions.

---

### Decision · Program_Chairs · 2026-03-14

**Decision:**

Accept (Oral)

**Comment:**

Dear Author(s),

On behalf of the Program Committee of the International Conference on Mathematics of Artificial Intelligence (MathAI 2026), we are pleased to inform you that your paper has been accepted for an oral presentation at MathAI 2026.

Your paper was evaluated through a rigorous two-stage review process involving both automated screening and expert review by members of the Program Committee. The reviewers recognized the quality and contribution of your work.

Presentation details:

- Format: Oral presentation (15–20 minutes + 5 minutes Q&A)
- Mode: You may present either in person (offline) at the conference venue in Sirius, Russia, or remotely via Zoom. Please indicate your preferred mode when confirming your participation.
- Conference dates: Marh 30 - April 3, 2026
- Website: https://mathai.club

Next steps:

1. Please confirm your participation and presentation mode by replying to this email mathai.club@yandex.ru no later than March 15, 2026 18:00 Moscow time.
2. If you plan to attend in person, the organizing committee will provide accommodation details separately.
3. Please prepare your final camera-ready manuscript according to the formatting guidelines available at https://mathai.club and upload it to OpenReview by March 15, 2026 18:00 Moscow time.

Should you have any questions regarding the program, logistics, or your presentation slot, please do not hesitate to contact us.

We look forward to your contribution to MathAI 2026.

With kind regards,

MathAI 2026 Program Committee
International Conference on Mathematics of Artificial Intelligence
https://mathai.club
OpenReview: https://openreview.net/group?id=mathai.club/MathAI/2026/Conference
Telegram: https://t.me/MathAI_club
Email: mathai.club@yandex.ru